# Aerobic Exercise Prevents Depression via Alleviating Hippocampus Injury in Chronic Stressed Depression Rats

**DOI:** 10.3390/brainsci11010009

**Published:** 2020-12-23

**Authors:** Jie Kang, Di Wang, Yongchang Duan, Lin Zhai, Lin Shi, Fei Guo

**Affiliations:** 1Physical Education Institute, Shaanxi Normal University, Xi’an 710119, China; kj1982@snnu.edu.cn (J.K.); wangd@snnu.edu.cn (D.W.); dyc123@snnu.edu.cn (Y.D.); zhailin@snnu.edu.cn (L.Z.); 2School of Food Engineering and Nutritional Science, Shaanxi Normal University, Xi’an 710119, China; 3Department of Biology and Biological Engineering, Division of Food and Nutrition Science, Chalmers University of Technology, SE-412 96 Göteborg, Sweden

**Keywords:** aerobic exercise, hippocampus injury, chronic stress, ionotropic glutamate receptors, depression related signaling molecules

## Abstract

(1) Background: Depression is one of the overwhelming public health problems. Alleviating hippocampus injury may prevent depression development. Herein, we established the chronic unpredictable mild stress (CUMS) model and aimed to investigate whether aerobic exercise (AE) could alleviate CUMS induced depression-like behaviors and hippocampus injury. (2) Methods: Forty-eight healthy male Sprague-Dawley rats (200 ± 20 g) were randomly divided into 4 groups (control, CUMS, CUMS + 7 days AE, CUMS + 14 days AE). Rats with AE treatments were subjected to 45 min treadmill per day. (3) Results: AE intervention significantly improved CUMS-induced depressive behaviors, e.g., running square numbers and immobility time assessed by the open field and forced swimming test, suppressed hippocampal neuron apoptosis, reduced levels of phosphorylation of NMDA receptor and homocysteine in hippocampus, as well as serum glucocorticoids, compared to the CUMS rats. In contrast, AE upregulated phosphorylation of AMPAR receptor and brain-derived neurotrophic factor (BDNF) hippocampus in CUMS depression rats. The 14 day-AE treatment exhibited better performance than 7 day-AE on the improvement of the hippocampal function. (4) Conclusion: AE might be an efficient strategy for prevention of CUMS-induced depression via ameliorating hippocampus functions. Underlying mechanisms may be related with glutamatergic system, the neurotoxic effects of homocysteine, and/or influences in glucocorticoids-BDNF expression interaction.

## 1. Introduction

Depression emerges as one of the overwhelming public health problems [1], and patients with depression increase at an annual rate of 11.3% worldwide [2]. Although pathologies of depression have not been fully elucidated, accumulating neuropsychological evidence has shown that chronic stress destructed balance between neuronal damage, and neuronal regeneration may contribute to atrophy of the hippocampal neurons, structural destruction, cell number decline, and hippocampal neuronal apoptosis, ultimately emerging clinical manifestations of depression [3,4,5]. Alleviation of hippocampal neurons injury or promotion of plasticity recovery after hippocampus injury may prevent the development of depression [6].

Over the last decade, studies have shown favorable effects of exercise on the treatment and rehabilitation of mental and cerebral diseases [7,8,9], considering exercise as a nonpharmacological and efficacious approach for the prevention and management of depression [4,8,10]. There have been many pieces of research on anti-depression effects of exercise in recent years; however, underlying mechanisms remain obscure [7,8,11,12]. Previous studies showed that aerobic exercise (AE) could stimulate neuron regeneration, regulate neurotrophin such as brain-derived neurotrophic factor (BDNF) and improve the plasticity of the brain [13,14]. Luo et al. [11] recently reported beneficial effects of AE on depression-like behaviors in chronic unpredictable mild stress (CUMS) mice and enhanced the expressions of p-AMPK and PGC-1alpha, the ratio of p-AMPK/AMPK, and boosted ATP content. Consistently, our recent study also reported the benefits of 8 week AE on improvement of CUMS–induced depressive behavior, neuron injury, and synaptic plasticity [15]. Of note, we found that AE helped to maintain normal amplitude of population spike and fEPSP slope in CUMS rats. This finding suggests effects of AE on preventing against long-term potentiation damage caused by chronic stress.

Depression development is not only associated with neurotransmitters and their receptors in the brain, but also involves into post-receptor signal transduction systems and gene transcription processes associated with neurotrophic disorders and reduction of neuroplasticity in specific brain regions [11,16,17,18]. Specifically, glutamatergic abnormalities have recently been implicated in the pathophysiology of depression. The ionotropic glutamate receptors (iGluRs), including AMPA receptor and Nmethyl-D-aspartate (NMDA) receptors, may play important roles [19,20,21]. Moreover, an increased level of circulating homocysteine (HCY), a key intermediate product in the metabolism of cysteine, has been associated with depression in different populations [22,23,24]. Previous studies also reported adverse effects of HCY on the hippocampus [25,26]. Interestingly, HCY may play a key role in neurotoxicity through activation of NMDA receptors-mediated signaling pathway [27]. In addition, glucocorticoids (GC) serve as key stress response hormones that are known to exacerbate neuronal injury, reduce hippocampal neurogenesis and to impair synaptic plasticity [28]. However, whether AE exerts antidepressant effects through AMPAR and NMDAR or other essential signaling molecules such as HCY and GC, has unfortunately not been sufficiently explored.

In the present study, we aimed to investigate impacts of AE during chronic stress procedure on depressive behaviors, hippocampal neuronal apoptosis, expression of iGluRs, BDNF and HCY expression in hippocampus, as well as serum GC. We also designed AE regimen for 7 days and for 14 days to examine time-effect on anti-depression performance of AE in CUMS rats.

## 2. Materials and Methods

### 2.1. Animal Experiments

Healthy male Sprague-Dawley rats (200 ± 20 g, *n* = 48) were obtained from the Laboratory Animal Breeding and Research Center of Xi’an Jiaotong University (Xi’an, China). The animals were maintained in a room with 12/12 h of light/dark cycle, in an air-conditioned room and had free access to water and diet. The experimental design is shown in Figure 1. Rats were randomly divided into control group (Control, *n* = 12), chronic unpredictable mild stress group (CUMS, *n* = 12), chronic unpredictable mild stress + 7 days of aerobic exercise group (CUMS + 7 AE, *n* = 12), and chronic unpredictable mild stress + 14 days of aerobic exercise group (CUMS + 14 AE, *n* = 12). Rats in the CUMS group were exposed to different stressors each day for 4 weeks to establish the CUMS model. Each rat was kept in one cage. The stress exposures were conducted in another room to avoid stress influences on the control group. The procedure of establishing CUMS-induced depression models was performed according to previous studies with minor modifications [15,29]. A variety of stressors were used, such as food deprivation for 24 h, water deprivation for 24 h, tail pinching for 1 min, overnight illumination, physical restraint for 2 h (plastic bottles), 45 °C hot water exposure for 5 min, 5 °C cold water exposure for 5 min, soiled cage exposure for 24 h (200 mL of water and 100 g of sawdust combined together), and 45 °C cage tilting for 7 h.

For CUMS + 7 AE and CUMS + 14 AE, exercise was intervened during the CUMS establishment (Figure 1), after the 1st week of stress paradigm, to assess its potential effects on preventing and alleviating CUMS-induced abnormalities in behaviors and hippocampus. Exercise and stress were performed separately and the resting interval between the two procedures was 2–3 h. Rats in the control group did not receive exercise or stressors, and were daily handled for 21 days.

The care with the animals followed the official governmental guidelines in accordance with the NIH Guide for the Care and Use of Laboratory Animals. This study was approved by the Ethics Committee for the use of animals of the Xi’an Jiaotong University Health Science Center (Xi’an, China).

### 2.2. Protocal of Aerobic Exercise

The AE protocol has been described by Fahey et al. [30] with slight modifications. Rats were trained on the treadmill (Anhui Zhenghua Biologicapparatus Facilities Co., LTD, Anhui, China). Each rat was placed on the separate belt of the treadmill running at a slow pace (0.78 km/h) for 5 min, and then the belt speed was increased to 1.02 km/h for 45 min. The exercise was conducted at 8:30–12:30 a.m. for 1 week (CUMS + 7 AE) or for 2 weeks (CUMS + 14 AE). Rats were forced to keep running during the exercise period by the electric shock stimulus (0.0–5.0 mA) implemented at the end of the belt. The open field and forced swimming test were conducted within 3 days after intervention regimen.

### 2.3. Behavioral Tests

#### 2.3.1. Open Field Test

The open field test was conducted using the method described by Arteni et al. [31]. Briefly, the floor of a wooden box with 100 cm × 100 cm × 20 cm was divided into 25 equal squares. The squares connected to the wall were outer squares, and other squares were central squares. The box was maintained dark light. Animals were placed on central squares and observed the running square numbers (times·(5 min)^−1^) and upright numbers (times·(5 min)^−1^) in rats.

#### 2.3.2. Forced Swimming Test

Details of forced swimming test (FST) were described by Slattery and Cryan [15,32]. Briefly, rats were forced to swim individually in clear Plexiglas cylinders (40 cm height, 18 cm diameter) that were filled with water (22~24 °C 30 cm depth). FST contained two parts: an initial 15 min pretest, followed 24 h later by a 5 min swimming test. Following each swim session, the rats were towel dried and returned to their home cages. The cumulative time spent in immobile posturing, i.e., minimal effort to keep head above water during the 5 min test, was recorded. The up time, i.e., struggling with limbs on a sidewalk in an attempt to escape the container during the 5 min test, was recorded.

### 2.4. Assessment of Hippocampal Neuronal Apoptosis

All rats were sacrificed after the end of the behavioral assessments by an overdose sodium pentobarbital (50 mg/kg body weight). The entire hippocampus of each brain was quickly collected on dry ice and stored at −70 °C until assayed. The hippocampus of rats in each group (6 per group) was used for detection of hippocampal neuronal apoptosis. Detailed protocol has been published elsewhere [33,34]. In brief, the hippocampus samples were collected and submerged in ice-cold Hank’s Buffered Salt Solution (Gibco, Carlsbad, CA, USA) for 5 min and were then minced with ophthalmic scissors and treated with 0.125% trypsin-0.02% EDTA for 15 min. Digestion was terminated by adding Dulbecco’s modified Eagle medium/F12 (DMEM/F12, Gibco, Carlsbad, CA, USA) with 10% fetal bovine serum (Gibco, Carlsbad, CA, USA). The sample was sequentially filtered through a 70-μm cell strainer and collected by centrifugation for 3 min at 1000 rpm. The precipitate was resuspended in neurobasal medium supplemented with Neurobasal-A+2%B27+0.5mM glutamine (Gibco, Carlsbad, CA, USA). Cells were attainted and plates were maintained in a 5% CO_2_ incubator at 37 °C. The medium was substituted for fresh DMEM after 24 h and then every three days.

Cells were collected after trypsinization and were rinsed with cold phosphate buffer saline for two times at 4 °C and centrifugation at 1000 rpm for 5 min). Cells were attainted at a final density of 1 × 10^10^/L. Apoptosis was assessed by using the Annexin V-FITC/PI doubling staining assay. After resuspension with 500 μL 1 × binding buffer, the mixture was incubated with 5 μL Annexin V-FITC in the ice bath for 10 min in the dark and was then mixed with 5 μL propidium iodide (10 mg/L) prior to analysis. Fluorescence levels were quantified by the flow cytometry (FACS-Aria, Becton Dickson, San Jose, CA, USA) [34,35].

The apoptosis rate was estimated by summarizing the apoptosis rate of early and late apoptotic cells detected by the flow cytometry. Annexin V-positive and propidium iodide negative cells were considered as early apoptosis cells, whereas annexin V positive and PI positive were regarded as late apoptotic cells.

### 2.5. ELISA Analysis

The hippocampus of remaining rats in each group (*n* = 6) was subjected to ELISA analysis. The hippocampus was homogenized with pre-cooling 0.1 M phosphate buffer (pH 7.4) 10 times (w/v). The homogenate was centrifuged at 12.000 rpm (4 °C) for 15 min to remove precipitation, and the supernatant were collected for analyses. The concentrations of phosphorylation of AMPA receptor (p-AMPA, JL15686, Shanghai Jianglai Biology, Shanghai, China), p-NMDA receptor (BJ-elisa-2559, Shanghai Bangjing Industry, Shanghai, China.), HCY (YM-AS1829, Shanghai Yuanmu Biology, Shanghai, China), and BDNF (RAB1138-1KT, Sigma, St. Louis, MO, USA) in the supernatant were measured using ELISA kits.

Blood samples were collected from the abdominal aorta and were kept in a water bath (37 °C) for 30 min. The serum was obtained by centrifugation (12,000× *g*, 20 min) and was then stored under −80 °C until analyses. The serum GC was measured using ELISA kits (yb-E12318, BioLegend, San Diego, CA, USA.).

### 2.6. Statistical Analyses

Data were expressed as mean ± standard deviation and were analyzed by one-way analysis of variance (ANOVA) followed by least significant difference multiple comparisons using the Statistical Package for the Social Sciences (SPSS 21.0, Chicago, IL, USA). A value of *p* < 0.05 was considered significant. Figures presenting differences in behavioral assessments and molecular alterations between groups were produced by Graphpad prism 7.0. Heatmaps showing correlations between behavioral and molecular alterations were drawn using MetaboAnalyst (www.metaboanalyst.ca).

## 3. Results

### 3.1. AE Improved Behavioral Indexes in CUMS-Induced Rats

CUMS increased running square numbers while lowered upright numbers (*p* < 0.001), compared with the control group (Figure 2). AE for 7 days significantly decreased CUMS-induced running square numbers (*p* < 0.001) and improved CUMS-induced upright numbers (*p* < 0.001). Similarly, AE for 7 days reversed the CUMS-induced increase of immobility time and reduction of up time in FST test (*p* < 0.001). Moreover, AE for 14 days during CUMS showed more pronounced effects than AE for 7 days in improving depressive behaviors (Figure 2).

### 3.2. AE Reduced CUMS-Induced Hippocampal Neuronal Apoptosis

The apoptosis rare of hippocampal neuron in CUMS group was significantly higher than in the control group (*p* < 0.001, Figure 3). AE for 7 days ameliorated CUMS-induced hippocampal neuron apoptosis (*p* < 0.01). The effect of AE for 14 days on the apoptosis rare of hippocampal neuron was significantly improved compared with AE for 7 days (*p* < 0.05).

### 3.3. AE Beneficially Regulated Levels of Ionotropic Glutamate Receptors in CUMS-Induced Depression Rats

To investigate the effect of AE on depression-related iGluRs, we measured levels of p-AMPA receptor and p-NMDA receptor that were involved in depression regulation in the hippocampus. CUMS reduced p-AMPA level in hippocampus (Figure 4A) while increased p-NMDA level (Figure 4B) was compared with the control group (*p* < 0.001). AE for 7 days enhanced CUMS-induced p-AMPAR level (*p* < 0.01) while it prevented an increase of CUMS-induced p-NMDAR level (*p* < 0.01). Consistently, AE for 14 days showed better performance than AE for 7 days in regulating CUMS-induced ionotropic glutamate receptors (*p* < 0.05).

### 3.4. AE Beneficially Regulated Levels of HCY, BDNF and GC in CUMS-Induced Depression Rats

To investigate the effect of AE on depression-related signal molecules, we measured levels of HCY and BDNF in hippocampus, and GC in serum (Figure 5). Compared with the control group, we observed substantial increases in HCY level in hippocampus and serum GC in the CUMS group (*p* < 0.001). On the contrary, the CUMS group had the lowest level of BDNF in the hippocampus (*p* < 0.001). AE treatment significantly reversed the CUMS-induced increase of hippocampal level of HCY and serum GC, while enhanced BDNF expression in hippocampus compared with the CUMS group. Moreover, the 14-day AE treatment exhibited better performance on regulating levels of depression-related signal molecules compared with the 7-days AE treatment (*p* < 0.05).

### 3.5. Correlations

We found that samples from the same treatment clustered perfectly by integrating behaviors indices and levels of key molecules involved in pathways related with pathogenesis of depression, using the hierarchical clustering analysis (Figure 6A). Specifically, a clear discrepancy was observed between CUMS group and control group. Of note, AE group beneficially improved CUMS-induced depressive behaviors and key molecules, in a training time-dependent manner. Moreover, strong correlations between behavioral indices and molecular alterations were observed (Figure 6B). These findings indicate that benefits of AE on alleviating CUMS-induced behaviors were accomplished by its favorable regulation of hippocampal neuronal apoptosis, expression of iGluRs, i.e., phosphorylation of AMPA receptor and phosphorylation of NMDA receptor, BDNF and HCY expression in hippocampus, as well as serum level of GC.

## 4. Discussion

Exercise has been consistently recognized for its therapeutic effects on depression. We found that AE significantly alleviated CUMS-induced depressive behaviors, suppressed hippocampal neuronal apoptosis, favorably regulated expression of iGluRs, and levels of depression-related signal molecules. Moreover, 14 day-AE regimen during chronic stress paradigm exhibited better anti-depression performance than the 7 day-AE, which have rarely been explored before. Of note, we found that benefits of AE on alleviating CUMS-induced behaviors were accomplished by its favorable regulation of hippocampal neuronal apoptosis, expression of iGluRs, BDNF and HCY in hippocampus, and serum GC. Our findings suggest that mechanisms underlying benefits of AE may include regulation of glutamatergic system, the neurotoxic effects of HCY in regulating ionotropic glutamate receptors, and/or influences in the glucocorticoids-BDNF expression interaction.

Recent studies have shown that AE improved depression-like behaviors or depressive symptoms in CUMS animal models [4,11,36]. Ryan et al. [37] found that an 8-week moderate-intensity aerobic exercise training intervention reduced depressive symptoms among individuals with major depressive disorder. In the present study, AE intervention was performed during the CUMS establishment. The findings demonstrate the effects of AE on preventing and alleviating of CUMS-induced depression like behaviors. Yet, the pathogenesis of depression and potential mechanisms underlying benefits of exercise on preventing depression are still poorly understood.

The hippocampus is the main encephalic region associated with the occurrence of depression [38,39]. Chronic stress has shown to reduce the hippocampal volume and detrimentally influence the morphological structure of neuron dendrites in animal studies [40]. Exposure to stress decreased neuroplasticity by increasing apoptosis and decreasing neurogenesis, and consequently resulted in depression [12]. Hippocampal neuronal apoptosis accelerates progression of depression [38,41]. Besides, our recent study showed that a short-term resistance training or AE alleviated CUMS induced apoptosis rate of hippocampal neuron [15]. Similarly, Chung et al. reported that treadmill exercise inhibited hippocampal apoptosis possibly through enhancing NMDA receptor expression in schizophrenic mice [42]. Seo et al. also found that aerobic exercise could effectively reduce stress-induced hippocampal apoptosis in PTSD animal model [12].

In the present study, we found that AE significantly ameliorated CUMS-induced apoptosis rate of hippocampal neuron and 2-week AE showed more pronounced effects. Moreover, AE significantly reduced expression of CUMS-induced an over-activation of p-NMDA receptors and increased the expression of p-AMPA receptor in hippocampus. Pharmacological evidence suggests that abnormal glutamate neurotransmission may be associated with the pathophysiology of mood disorders [43,44]. Several studies have shown that occurrence of stress-induced depression was associated with expressions of ionotropic glutamate receptors, i.e., NMDA receptor and AMPA receptor, a key molecule for synaptic efficacy [21,45,46,47]. These receptors have been involved in many physiological and pathological brain conditions. Stress-caused excessive increase of glutamate in the hippocampus and over-activation of NMDA receptors have been considered important causes of stress-induced depression [48]. Clinical evidence showed that the expression of AMPA receptor was significantly reduced in the brain of depression patients [45]. These findings support that preventing hippocampal neurons from being damaged, reducing the degree of damage, or promoting recovery after injury, may be possible mechanisms of antidepressant effects of exercise [16,49].

Moreover, we observed that AE reduced CUMS-induced HCY level in hippocampus. Homocysteine is a metabolite of the methionine and has been reported to play an important role in neurotoxicity through activation of NMDA receptor-mediated signaling pathway [27,50,51]. Hyperhomocysteinemia was associated with the development of neurological and psychiatric diseases, including depression [51,52,53]. In a recent study, Poddar et al., identified AMPA receptors as novel intermediators involved in a crosstalk between extracellular signal-regulated kinase and p38 MAPKs, which is a well-known signaling pathway of homocysteine-induced neurotoxicity [27]. Taken together, our findings support beneficial impacts of AE on ameliorating hippocampus injury through regulating the glutamatergic system [43]. The neurotoxic role of HCY in regulating ionotropic glutamate receptors involved in the pathogenesis of depression has also been highlighted.

Furthermore, we found that short-term AE intervention significantly prevented the CUMS-increased BDNF expression in the hippocampus, in line with previous studies [54,55]. Increased BDNF expression has been associated with neuroplasticity impairment, one of the pathogenesis of depression. Although potential mechanism by which exercise enhances BDNF expression is not fully understood, previous animal studies have shown that stress-induced reduction of hippocampal BDNF expression was associated with increased serum GC level [56,57,58], in agreement with our findings. The glucocorticoid receptor in the hypothalamic–pituitary–adrenal axis has been consistently shown to play a pivotal role in the negative feedback regulation of GC level in the blood, which was putatively involved in the onset of depression when its concentration was abnormally high [58]. The dysfunction of the hypothalamic–pituitary–adrenal axis may adversely influence BDNF function and consequently cause depressive behaviors [58]. Our study suggests that AE may prevent depression through beneficially affecting glucocorticoids, thereby leading to an increase in BDNF expression in the hippocampus. However, it should be noted that the exercise induced increase in BDNF expression has also been associated with neurotransmitters such as norepinephrine and 5-hydroxytryptamine (5-HT) [59]. This indicates that complex and multiple mechanisms may contribute to effects of exercise on up-regulating BDNF expression.

Our study has several limitations. First, chronic unpredictable mild stressors often result in anhedonic and resilient rats. The core symptom of depression-like behavior, a decrease in sucrose preference, should have been assessed to evaluate models for better characterization of the depressive animals. Instead, we only used the open field test and forced swimming test to assess CUMS-induced depression-like behaviors. However, it is noteworthy that we did not select animals for further studies based on behavioral tests. The reliability of findings regarding hippocampal neuronal apoptosis, brain-derived neurotrophic factors and serum level of glucocorticoids may not be influenced. Second, we failed to investigate whether exercise could exhibit a significant effect on hippocampal markers, no matter with or without CUMS. Herein, we focused on changes in stress-induced markers in the rat hippocampus between CUMS, CUMS + 7 AE, and CUMS + 14 AE, revealing improvements of exercise on alleviating depression induced dysfunctions. Particularly, comparing AE regimen for 7 days and for 14 days is one of the strengths to examine time-effect on anti-depression performance of AE in CUMS rats, which have been rarely evaluated. Whereas, it should be noted that, according to our study design, the time from the last day of CUMS + 7 AE to sacrifice was longer than for CUMS + 14 AE group, which may affect the obtained results regarding time-effect on anti-depression performance of AE. Another strategy would be that starting CUMS + 7 AE at 3rd week to avoid such time differences. But, in this case, rats for CUMS 7 AE groups would have worse physical and psychological conditions, leading to the bias for grouping animals.

## 5. Conclusions

Our study showed healthy impacts of the short-time AE on prevention of CUMS-induced behavioral disorder and hippocampal neuronal apoptosis. AE beneficially altered levels of depression-related iGluRs, i.e., p-AMPA receptor, p-NMDA receptor, and other essential signaling molecules including BDNF, HCY in hippocampus and GC in serum. Of note, we found that AE for 14 days during CUMS greatly exhibited better performance than AE for 7 days on depression prevention. Moreover, strong correlations between behavioral indices and molecular alterations regulated by AE regimen were observed. The observed benefits of AE may be mechanistically explained by regulation of glutamatergic system, the neurotoxic effects of HCY in regulating ionotropic glutamate receptors involved in the pathogenesis of depression, and/or influences in glucocorticoids-BDNF expression interaction.

## Figures and Tables

**Figure 1 brainsci-11-00009-f001:**
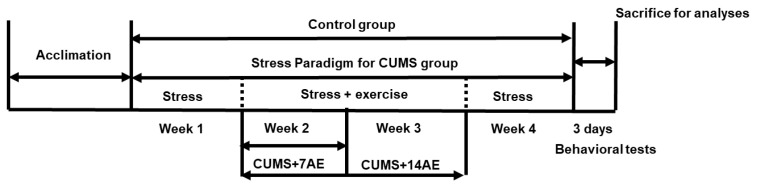
Experimental design. Open field test and forced swimming test were performed on after 4 weeks intervention. CUMS, chronic unpredictable mild stress.

**Figure 2 brainsci-11-00009-f002:**
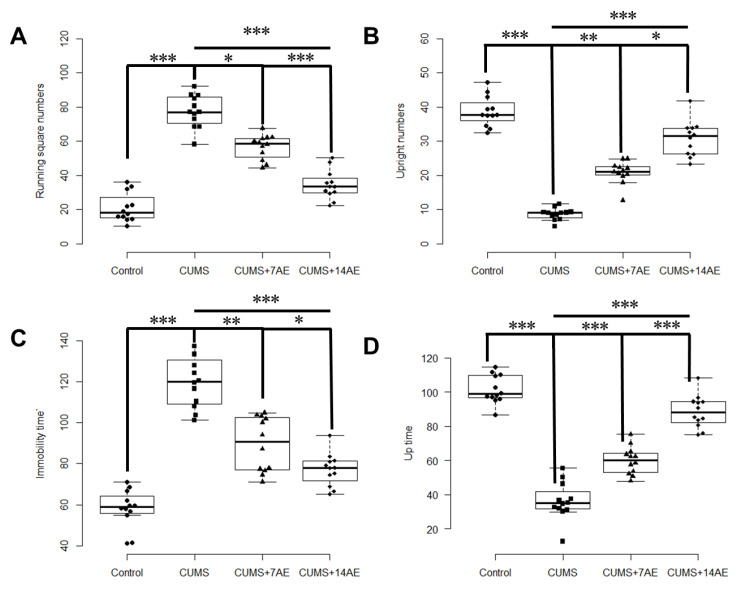
Effects of aerobic exercise on CUMS-induced behaviors, i.e., running square numbers (times·(5 min)^−1^) (**A**), upright numbers (times·(5 min)^−1^) (**B**), immobility time (times·(5 min)^−1^) (**C**), and Up time (times·(5 min)^−1^) (**D**). * *p* < 0.05; ** *p* < 0.01; *** *p* < 0.001; CUMS: chronic unpredictable mild stress; 7AE: Aerobic exercise for 7 days; 14AE: Aerobic exercise for 14 days.

**Figure 3 brainsci-11-00009-f003:**
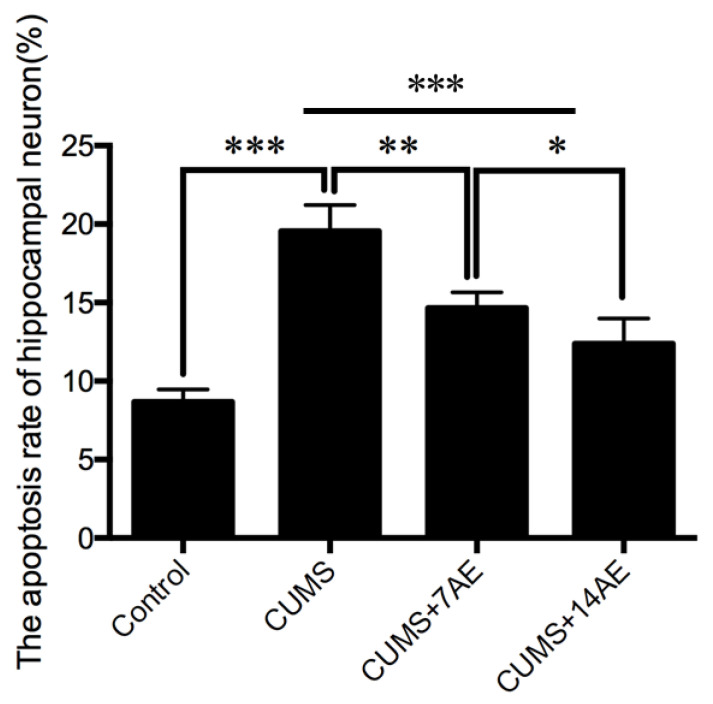
Effects of aerobic exercise on CUMS-induced hippocampal neuronal apoptosis. * *p* < 0.05; ** *p* < 0.01; *** *p* < 0.001; CUMS: chronic unpredictable mild stress; 7AE: Aerobic exercise for 7 days; 14AE: Aerobic exercise for 14 days.

**Figure 4 brainsci-11-00009-f004:**
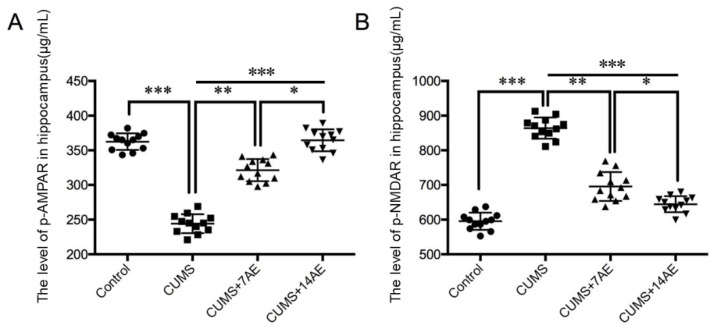
Effects of aerobic exercise on p-AMPAR (**A**) and p-NMDAR (**B**) in hippocampus of CUMS-induced depression rats. * *p* < 0.05; ** *p* < 0.01; *** *p* < 0.001; CUMS: chronic unpredictable mild stress; 7AE: Aerobic exercise for 7 days; 14AE: Aerobic exercise for 14 days.

**Figure 5 brainsci-11-00009-f005:**
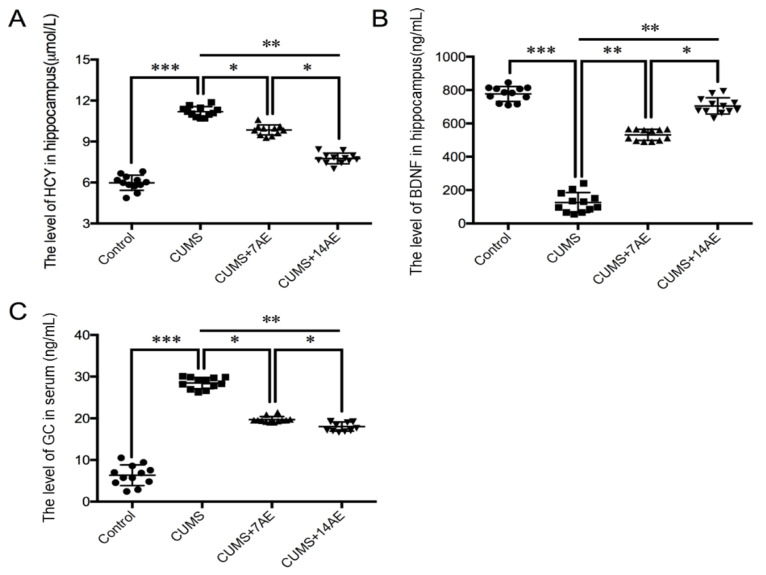
Effects of aerobic exercise on depression-related signal molecules. (**A**): homocysteine (HCY); (**B**): Bain-derived neurotrophic factor (BDNF); (**C**): serum glucocorticoids (GC) in CUMS-induced depression rats. * *p* < 0.05; ** *p* < 0.01; *** *p* < 0.001; CUMS: chronic unpredictable mild stress; 7AE: Aerobic exercise for 7 days; 14AE: Aerobic exercise for 14 days.

**Figure 6 brainsci-11-00009-f006:**
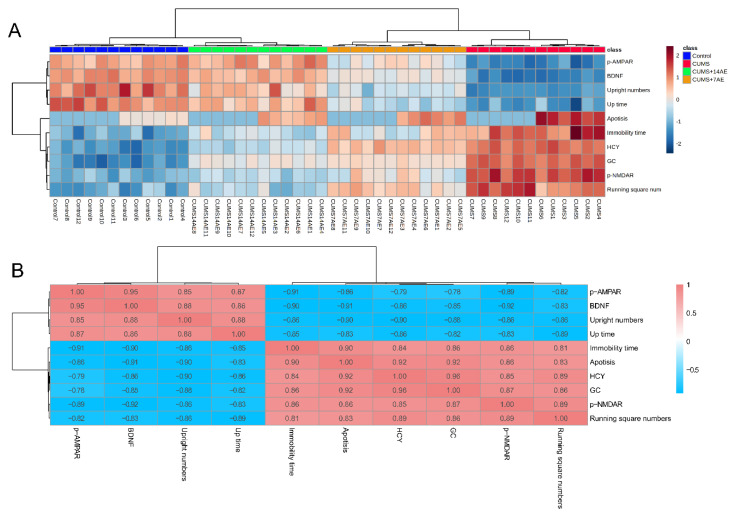
Clusters derived from behavioral indices and molecular alterations discriminating groups (**A**). Correlations between behavioral indices assessed by the open field test and forced swimming test, expression of iGluRs, i.e., phosphorylation of AMPA receptor and phosphorylation of NMDA receptor, brain-derived neurotrophic factor (BDNF), homocysteine (HCY) in hippocampus as well as serum glucocorticoids (GC) (**B**). CUMS: chronic unpredictable mild stress; 7AE: Aerobic exercise for 7 days; 14AE: Aerobic exercise for 14 days.

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
