# Peer review of "Aerobic Exercise Prevents Depression via Alleviating Hippocampus Injury in Chronic Stressed Depression Rats"

_brainsci, 2020, doi:10.3390/brainsci11010009_

Round 1

Reviewer 1 Report

Brain Sciences: Review
Kang. et al employed the chronic unpredictable mild stressors paradigm to explore aerobic exercise-induced mitigation in stress-induced markers in the rat hippocampus. The study design is not immaculate; however, the findings add significant information to the current knowledge.
There are a few major issues with the study design and data analysis.
1. Chronic unpredictable mild stressors result in anhedonic and resilient rats after the stress paradigm, however, the current study not mentioned this.
2. Why aerobic exercise was started just after 1 week of stress paradigm?
3. Additionally, one more group with the only exercise would also add significant information and would support the findings.
4. There is no correlation analysis between the molecular and behavioral data, which may provide further insights a plausible link between behavioral and molecular alterations.
5. What is the normal method to isolate hippocampal neurons of rats in the methodology section?
6. More details about the flow cytometry machine and methodology is required.
7. Blood and serum collection is also not clear?
8. Details of ELISA kits should be provided, such as catalog number. That would help to reproduce the findings by the other groups in the future.
9. How apoptosis rate was estimated in the hippocampus?
10. In the discussion section “Seo et al also found that aerobic exercise could effectively improve stress-induced hippocampal apoptosis in PTSD animal model”, which should be effectively reduced.
11. In general sentences are long and complex, I would suggest using short and clear sentences.

Reviewer 2 Report

The manuscript “Aerobic exercise prevents depression via alleviating hippocampus injury in chronic stressed depression rats” describes the molecular basis of depressive-like behavior in an animal model of stress-induced depression in rats. Chronic stress and anxiety are key factors for the development of neuropsychiatric diseases, therefore the proposed studies are in point of the current topic especially now when depressive episodes spiked during the COVID pandemic. Depressive behavior correlates with hippocampal atrophy, disturbed glutamatergic neurotransmission, and neurogenesis leading to cognitive deficits. The authors addressed the use of aerobic exercises with a battery of chronic unpredictable stressors to create a model of hippocampal recovery from neuronal injury caused by chronic stress. The presented studies aim to determine the impact of 1-week and 2-weeks of aerobic exercises during chronic stress procedure based on chronic unpredictable mild stress paradigm on development of depressive-like behavior (open field, forced swimming test), hippocampal apoptosis markers (flow cytometry), expression of iGluRs, BDNF, homocysteine in the hippocampus, and glucocorticoid level in serum blood (Elisa assays) of Sprague-Dawley rats. Stress procedure and behavioral evaluation of depressive-like behavior are generally properly chosen. However, a test to assess depressive behavior not based on physical condition is missing. Weekly and 2-weeks AE may affect the performance of the open field test and the forced swimming test regardless of symptoms of depression. Therefore, test such as a sucrose preference test would be beneficial for better characterization of animals. However, due to the fact that the selection of animals for further studies was not based on behavioral tests, this does not affect the reliability of the results obtained from tissues. The study could be better designed, but the results obtained are reliable and informative. The discussion is well articulated, however, the authors do not clearly discuss the significance and novelty of their own results.

Minor corrections:

  1. The methodology section „Assessment of hippocampal neuronal apoptosis” should be described in more details especially methods underlies flow cytometry analysis due to the lack of information about antibodies used in the experiments to detect apoptosis rate. Moreover, in 129 line authors wrote „according to the normal method to isolate hippocampal neuron of the rat” - “normal method” should be also explained more precisely by adding short description or references.
  2. In the protocol of aerobic exercise lack of information about animal housing (single? paired?) and experimental conditions of treadmill running (how the run distance was measured and calculated per animal, what was the procedure when rats don’t want to exercise in a treadmill?). Supplementary material has to be added to clarify the protocol applied.
  3. Results presented behavioral readout should be presented in scatter plots because in this form are not informative enough.
  4. In 129 line should be probably “1 × 1010/L” instead of “1 × 1010/L”
  5. In 3.1, 3.2. 3.3, 3.4 the authors don't compare CUMS to CUMS with AE for 14 days, why?
  6. The time from the last day of AE to sample collection is different than for animals from CUMS with AE for 14 days and CUMS with AE for 7 days groups, which is an additional factor that may affect the obtained results. Perhaps not only the length of AE, but also the time that has elapsed since their completion, affects the tested parameters - the authors ignore this possibility in the description of the results and the discussion.
